# Redesigning and validation of fertilizer use in maize for variable plant densities in central rift valley and Jimma in Ethiopia

**Workneh Bekere Kenea** [1]*, **Tesfaye Balemi**[2], **Amsalu Nebiyu**[1]

**1** Department of Horticulture and Plant Sciences, Jimma University, Jimma, Ethiopia, **2** Crop Production Specialist, Food and Agriculture Organization, Semera Field Office, Semera, Ethiopia

* keneni02@yahoo.com

## Abstract

Due to low adoption and sub-optimal fertilizer use and planting density recommendation in maize, redesigning and testing these technologies are required. The study was conducted to evaluate redesigned fertilizer use of maize in two pant densities (32,443 and 53,333 plants ha$^{-1}$ in Central Rift Valley (CRV); 27724 and 62,000 plants ha$^{-1}$ in Jimma) on farmers' fields in contrasting agro-ecologies of Ethiopia. The on-farm study was conducted in the 2017 and 2018 cropping seasons with 3 × 2 fertilizer and plant density, factors in both regions of Ethiopia. In redesigned fertilizer use, nutrients were estimated based on the target yield. In this study, 40.8, 0.0, and 12.2 kg ha$^{-1}$ N, P, and K were estimated for the redesigned fertilizer use in CRV (50% of water-limited potential yield (Yw) = 3.1 t ha$^{-1}$) whereas in Jimma (50% of Yw = 7.5 t ha$^{-1}$) 149.8, 9, 130.6 kg ha$^{-1}$ N, P and K were estimated to produce the 50% of Yw. Linear mixed modeling was used to assess the effect of fertilizer-plant density treatments on maize yield and nutrient use efficiency. The result revealed that the average estimated maize yield for WOF, FFU, and RDFU fertilizer treatments were 2.6, 3.6, and 4.5 t ha$^{-1}$ under current plant density (32,443 plants ha$^{-1}$) in CRV whereas the average yields of these treatments were 3.2, 4.5 and 4.5 t ha$^{-1}$ respectively when maize was grown with redesigned plant density (53,333 plants ha$^{-1}$) in the same location. The average maize yield with WOF, FFU, and RDFU were 3.0, 4.6, and 4.6 t ha$^{-1}$ with 27,774 plants ha$^{-1}$ plant density in Jimma whereas the average maize yields over the two seasons with the same treatments were 4.3, 6.0 and 8.0 t ha$^{-1}$ respectively when the crop is planted with 62,000 plants ha$^{-1}$ plant density. The RDFU and redesigned plant density resulted in significantly higher yield compared to their respective control CRV but RDFU significantly increased maize yield when it was planted at redesigned (62,000 plant ha$^{-1}$) in Jimma. FFU and RDFU were economically viable and redesigned plant density was also a cheaper means of improving maize productivity, especially in the Jimma region. Soil organic carbon and N were closely related to the grain yield response of maize compared to other soil factors. In conclusion, this investigation gives an insight into the importance of redesigned fertilizer use and redesigned plant density for improving maize productivity and thereby narrowing the yield gaps of the crop in high maize potential regions in Ethiopia like Jimma.

**Data Availability Statement:** All relevant data are within the paper and its Supporting Information files.

**Funding:** The funder of this study was Bill & Melinda Gates Foundation [INV-008260]. The

funder did not play any role in the study design, data collection and analysis, decision to publish, or preparation of the manuscript.

**Competing interests:** The authors declared no competing interest.

## Introduction

Food insecurity remains a major concern in sub-Saharan African (SSA) countries including Ethiopia. The projected grain demand in 2050 for the region is 70% more than the current [1] as the population in the region is expected to double [2]. This higher domestic demand can either be achieved by increasing production or importing from other countries. In Ethiopia, increasing productivity and thereby narrowing yield gaps is possible mostly through sustainable intensification. Sustainable intensification is increasing productivity by growing high-yielding crop varieties with appropriate agronomic practices in a season or multiple seasons within a year on the existing cropping area [3]. Despite the possibility of increasing production with an area expansion, it is argued that increasing production through area expansion is not advisable for two reasons [4, 5]. Firstly, there is no area for the expansion as suitable land is already under cultivation. Secondly, bringing more land under cultivation is associated with increasing greenhouse gas emissions to the atmosphere which exacerbates global warming [6, 7].

Maize plays an important role in contributing to food security in Ethiopia. It is the cheapest source of calories [8] and hence, the majority of smallholder farmers grow the crop for consumption. Improving the productivity of this crop is, therefore, addressing the food security constraints of a large number of people in the country. Despite the large maize production potential (favorable climate, diverse genotypes for most agroecologies, and well-drained soil) of the country, the current maize yield is far below the potential yield. These low yields have resulted from sub-optimal input use as a result of the economic constraint of the farms. Studies indicated that the cereal yield gap in Ethiopia is high and the country can be cereal self-sufficient by closing the yield gaps (to 80% of Yw) of the crop without import and area expansion [1]. The yield gap is the difference between the water-limited potential yield and the actual yield [9]. Significant progress has been made in maize production since 2001 in Ethiopia [8]. The recent report of the Central Statistical Agency (CSA) also showed that maize production is increasing both in cultivation area and productivity. Accordingly, 12% more area was cultivated with maize in 2018 compared to the area occupied by the crop in 2014 whereas, the national average yield of maize increased from 3.4 t ha-1 in 2014 to 4 t ha-1 in 2018 [10, 11] which is equivalent to 18% yield increase over the past four years.

The previous fertilizer and planting density recommendation in maize in Ethiopia are suboptimal and perhaps adoption by smallholder farmers is generally low. Moreover, research on maize management practices so far hardly addressed the variability of soil resources that exist in nature or that are created by farmers through activities over time. This biophysical variability needs different intervention approaches based on the types of constraints they are facing [12–14]. These call upon redesigning the current fertilizer use and planting density of the crop tailoring to the biophysical and socioeconomic conditions in fields within a given region. Redesigning is a configuration of production technologies based on the prevailing context in which the technologies are improved, tested, and verified in fields while taking the constraints and opportunities of farms into account. The approach of redesigning crop management practices in this study, therefore, focuses on redesigning easily accessible and applicable technologies by smallholder farmers such as redesigning density with the use of optimal nutrient requirements that lead to optimal nutrient use efficiency and sustainable crop production The optimal input requirements [15] are the quantity of nutrients (N, P, K, and others) that are removed from soil by the crop during the growing period.

The overall goal of our study was to understand whether variable plant densities of maize require different fertilizer uses. The specific objectives of the study were therefore to (1) redesign and test fertilizer use and plant density under on-farm conditions and unravel their effect on maize yield and yield gaps (2) understand the economic feasibility of redesigned fertilizer

use and redesigned plant density in maize production at smallholder level through on-farm experimentation.

## Materials and methods

### Characteristics of the study area

CRV is located in southeastern Ethiopia whereas Jimma is located in the southwestern part of the country. The soil texture is predominately sandy loam in the former region whereas the soil texture class in Jimma is clay in most fields. Daily rainfall, minimum and maximum temperature during maize growing seasons in 2017 and 2018 were presented for CRV and Jimma regions (Fig 1 and S1 File).

### Treatments (current and redesigned practices)

The average fertilizer use and average plant density of maize in the CRV and Jimma regions of Ethiopia are presented in Table 1. They were obtained from a farm survey conducted in collaboration between the International Maize and Wheat Improvement Center (CIMMYT) through the Taking Maize Agronomy to Scale in Africa (TAMASA) project and the Ethiopian Institute of Agricultural Research. The average fertilizer use of the farmers in the regions is

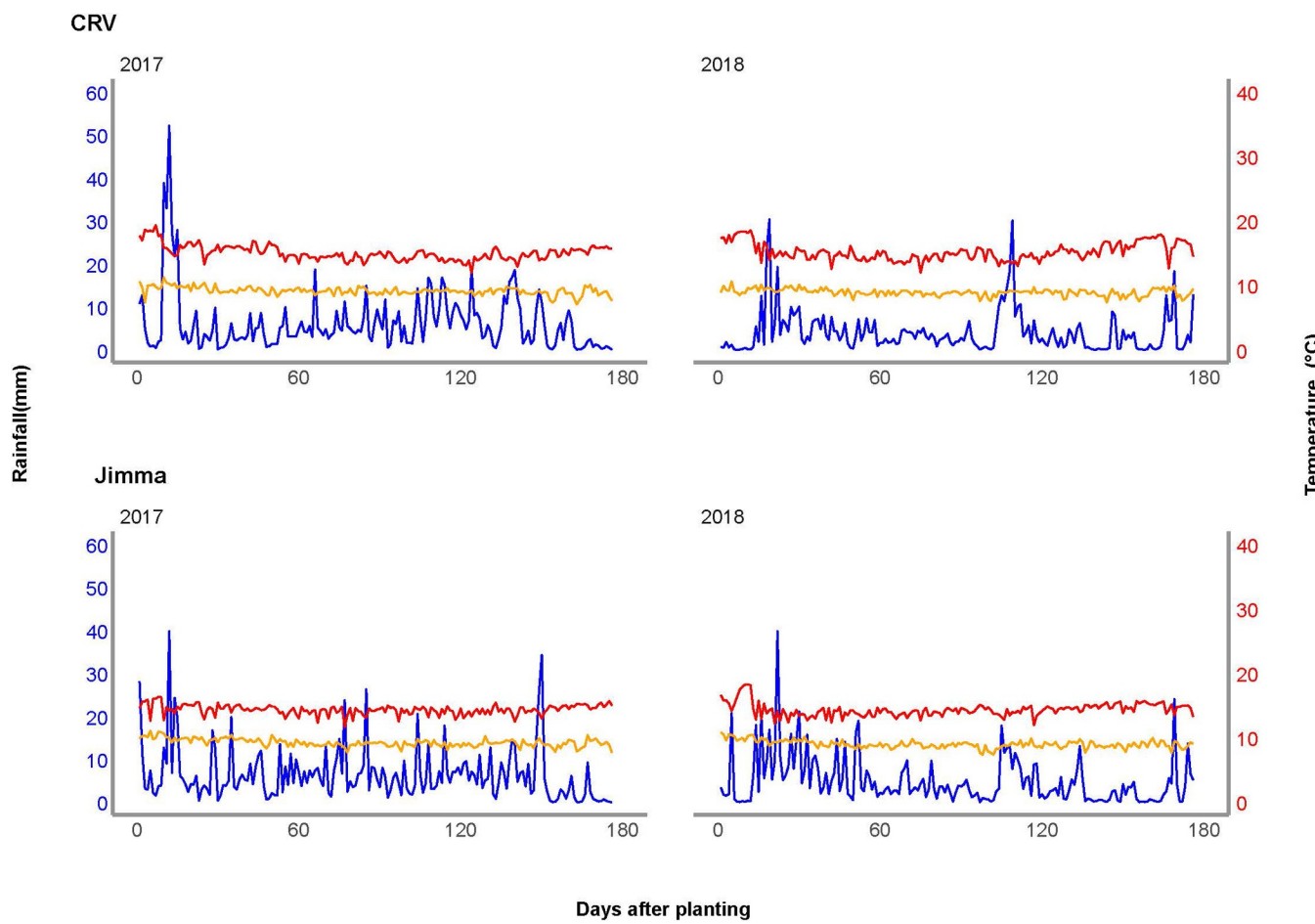

**Fig 1.** Daily rainfall (blue line on left axis), minimum temperature (orange on right axis) and maximum temperature (red line on right axis) for CRV and Jimma during maize growing period in 2017 and 2018 seasons.

**Table 1. Factorial combination of fertilizer use and plant density in CRV and Jimma.**

| Region | Levels | Factors | Values | | |
|---|---|---|---|---|---|
| | | **Nutrient management** | **N (kg/ha)** | **P (kg/ha)** | **K (kg/ha)** |
| CRV | 1 | 0 NPK | 0.0 | 0.0 | 0.0 |
| | 2 | Farmer's use | 21.5 | 12.8 | 0.0 |
| | 3 | Redesigned NPK (50% x Yw) | 40.8 | 0.0 | 12.2 |
| | | **Plant density (Plants ha$^{-1}$)** | | | |
| | A | Current | 32,443 | | |
| | B | Recommended | 53,333 | | |
| | | **Nutrient management** | N (kg/ha) | P (kg/ha) | K (kg/ha) |
| Jimma | 1 | 0 NPK | 0.0 | 0.0 | 0.0 |
| | 2 | Farmer's use | 53.2 | 30.0 | 0.0 |
| | 3 | Redesigned NPK (50% x Yw) | 149.8 | 9.0 | 130.6 |
| | | **Plant density (Plants ha$^{-1}$)** | | | |
| | A | Current | 27,724 | | |
| | B | Redesigned | 62,000 | | |

Values of nutrient amount and plant density were shown for both regions

farmer's fertilizer use (FFU) and current planting density is the average density used by farmers in the respective regions. The values of the current and redesigned fertilizer use, as well as current and redesigned densities in both regions, are shown in Table 1. From the combination of 3 levels of fertilizer use, 2 levels of plant densities (3× 2), there were 6 treatments in each region (see Table 1 for the detail).

## Principles in redesigned fertilizer use and redesigned plant density

In redesigned fertilizer use, N, P, and K nutrients were estimated based on the target yield to be produced [16, 17]. The target yield was set at 50% of Yw for both seasons. The target yields were obtained from Global Yield Gap and Water Productivity Atlas [18]. The ratio of target yield to the medium physiological use efficiency of the crop was computed. This ratio is the nutrient uptake requirement of the crop for achieving the target yield [19]. However, there are two sources for plant nutrient uptake: (1) indigenous soil supply and (2) fertilizer input. The soil-supplied nutrients (N, P, and K) were obtained from a nutrient omission trial [19] and were subtracted from the total uptake. The remaining uptake was supplied only from the fertilizer input. Lastly, these nutrients were corrected for their recovery fractions. The recovery fraction of nutrients is estimated from the ratio between uptake and applied amount. For N, P, and K, the average recovery efficiencies in tropical soils were 0.5, 0.1, and 0.5 respectively. The basis of redesigned plant density (CRV = 53,333 plants ha$^{-1}$ and 62,000 plants ha$^{-1}$) is the performances in previous demonstration trials and consultation of senior agronomists. Soil resources and climatic elements (rainfall) of the regions were taken into account. These densities were believed to be the best in the respective region in terms of yield improvement and sustainable maize production.

## Experimental setup

Farms that were involved in the farm survey of TAMASA hosted the experiment. In this study, farms represent smallholder farmers with their farm resources and farm management strategies. In both regions, 10 farms were selected in each season, in 2017and 2018 main growing seasons. The treatments were replicated over farms and hence one farm was a replicate. Fields

were prepared with oxen-drawn implements by farmers in both regions. The plot sizes in CRV and Jimma were 6 m × 6 m and 6.0 m × 6.4 m respectively. The spacing of current density (32,443 plants ha$^{-1}$) in CRV was 0.75 m between rows and 0.40 m between plants. For the redesigned density (53,333 plants ha$^{-1}$) in this region, the spacing was 0.75 m between rows and 0.25 m between plants. In Jimma, the spacing was 0.8 m between rows and 0.45 m between seeds for the current density (27,724 plants ha$^{-1}$). For the redesigned density (62,000 plants ha$^{-1}$), 0.8 m between rows and 0.2 m between plants were used in 2017. However, it was recognized that plants within a row were close to each other and exacerbated lodging in some fields in 2018. A discussion was made with farmers on how to resolve the planting orientation and most farmers suggested reducing row distance from 0.8 m to 0.5 m and increasing plant-to-plant distance to 0.25 m. Based on this feedback, 0.5 m by 0.25 m spacing was used in Jimma for 62,000 plant ha$^{-1}$ in 2018. For the study, the BH-540 variety which is commonly grown in CRV was used whereas BH-661 which is commonly grown in western Ethiopia was used in Jimma. The maturity duration of BH-660 is 160 days whereas the maturity duration of BH-540 is 140 days. Planting was done at the time when enough moisture was stored in the soil [20] in the third week of May in CRV and in the last week of May in Jimma. In 30–45 days after emergency, two seeds were planted at sowing in the same spot and one seedling was removed after crop establishment to maintain the intended plant population. Harvesting was done from net plot size excluding the two border rows. The net plots were 6m × 4.8m in Jimma whereas in CRV the net plot size was 6m × 4.5m. Harvesting time was at physiological maturity when grain moisture was dropped to 15–30% in most cases. This can be visually observed as leaves of the crop dries, cobs turns downwards from upright position.

## Statistical analysis

To assess the effect of fertilizer use on maize yield, linear mixed modeling was conducted using lme4 package [21]. The effects of the fertilizer treatments were expressed as derivatives of the baseline (farmer's fertilizer use). In the model, maize yield was used as response variable whereas fertilizer use, season number and plant density were used fixed factors. To address yield variability due to farms, the farms were used as random factor. Two-step models were developed (1) fertilizer treatments as fixed factor and farms as random factor (2) fertilizer treatment, and plant density as fixed factors and farms as random factor. The better fit model was assessed by comparing of the AIC of the two models with the associated P value. Significant effects of model parameters on grain yield were evaluated using the 'lmerTest' package in R software [22].

## Internal nutrient use efficiency

Grain and straw samples were collected from the on-farm trial of the first (2017) season. The N, P, and K concentrations in grain and straw yields were analyzed at Wageningen University and Research (WUR). Grain and straw samples were collected from redesigned plant density and dried to constant weight and then grinded to a size of less than 5 mm mesh size (to make fine powder) at the soil and plant analysis laboratory of Melkassa Agricultural Research Center in Ethiopia and sent to WUR, the Netherlands. The samples were digested with concentrated $H_2SO_4$ at elevated temperature (330˚C) using selenium (Se) as a catalyst. Salicylic acid was added to prevent the loss of nitrate-N. Total N and P were measured spectro-photometrically with a segmented-flow system (Skalar San++ System). In the same digests, K was measured with Varian AA240FS fast sequential atomic absorption spectrometer (Terneuzen, the Netherlands). The measured N, P, and K were converted to kg per hectare based on the obtained

grain and straw yields from the same treatments. The internal use efficiency of the nutrients was computed from the ratio of economic yield to nutrient uptake [23].

## Economic feasibility of the practices

The economic feasibility of recommended and redesigned plant densities (53,333 plants ha$^{-1}$ in CRV; 62,000 plants ha$^{-1}$ in Jimma) was assessed by computing the kilogram of extra seeds needed for planting these densities compared to the kilogram of seeds used for planting current density (32,443 plants ha$^{-1}$ in CRV 27,724 plants ha$^{-1}$ in Jimma). To obtain the cost of extra seed, the amount of extra seed was multiplied by the seed cost. The cost of 1 package (12.5 kg) of BH-540 seed was 370 ETB which is equivalent to 30 ETB/kg, whereas the cost of 1 package (12.5 kg) of BH-661 seed was 334 ETB which is 27 ETB/kg of seeds (price at Ano agro-industry). Maize price in CRV was 5 ETB kg$^{-1}$ in 2017 and 7.5 ETB kg$^{-1}$ in 2018. The transporting cost of fertilizer from the nearest union to home and maize grain to the nearest local market in CRV was 5 ETB 100 kg$^{-1}$. The cost of fertilizers in current and redesigned fertilizer uses was calculated based on the purchasing cost of the fertilizers obtained from farmers' unions in the respective regions. To calculate the value-cost ratio, the maize yield difference between fertilizer uses and the yield obtained from unfertilized treatment was computed. This yield difference was multiplied by its farm gate price. Then, the value cost ratio was computed from the ratio of marginal return of extra maize to total fertilizer (fertilizer cost, fertilizer transportation from input dealer to farmers' homes, and maize transportation from home to market) cost for both current and redesigned fertilizer uses. The average cost of transporting 100 kg of fertilizer from the input seller to the home and transporting maize from the home to the nearest market was 10 ETB per 100 kg of maize. The average maize grain price in Jimma was 12 ETB kg$^{-1}$ of maize grain in both 2017 and 2018. If the value-cost ratio is greater than 1, then using the production practices (planting density and fertilizer uses) is feasible for maize production whereas if the ratio was less than or equal to 1, the production practices are not economically feasible.

## Relation of maize yield and soil factors

Principal component analysis (PCA) was conducted to assess the relation of maize grain yield with soil properties of the maize field experiment. The maize grain yield was an average experimental yield of 2017 and 2018 seasons grown under NPK (redesigned fertilizer use). We assume that there was no yield limitation from nutrient perspective under NPK (redesigned fertilizer use). The soil properties used in this analysis were pH, organic carbon (OC), soil N content, soil P content, and exchangeable potassium. The factoextra R package was used for PCA analysis [24].

## Result

### The effect of fertilizer use on maize yield

The average estimated maize grain yield for farmer's fertilizer use (FFU) over two seasons in CRV was 5.2 t ha$^{-1}$ whereas it was 4.6 t ha$^{-1}$ for the FFU in Jimma. The yield increase (0.48 t ha$^{-1}$) due to RDFU in CRV was not significant whereas the yield increase (1.42 t ha$^{-1}$) due to RDFU treatment in Jimma was significant (p = 0.03). The interaction of FFU and season number was significant in both regions (p = $1.8 \times 10^{-8}$ in CRV and P = 0.04 in Jimma) whereas the interaction of RDFU with season number and WOF with season number was not significant in both regions (Table 2). This shows that the yield of FFU varied over season whereas the yield of RDFU and WOF treatments was not affected by season number and were stable over

**Table 2. Mixed model parameter estimates with standard errors in brackets of fertilizer use over two seasons of maize grain yield (t ha$^{-1}$) at 15% moisture content in CRV and Jimma, Ethiopia.**

| Region | parameter | Estimates (t/ha) | Stand Error | P value |
|---|---|---|---|---|
| CRV | Intercept (FFU) | 5.21 | 0.33 | $2 \times 10^{-16}$ |
| | RDFU | 0.48 | 0.40 | 0.23 |
| | WOF | -1.20 | 0.40 | 0.003 |
| | FFU × SN | -2.34 | 0.40 | $1.8 \times 10^{-8}$ |
| | RDRF × SN | -0.75 | 0.57 | 0.18 |
| | WOF × SN | 0.11 | 0.57 | 0.83 |
| Jimma | Intercept (FFU) | 4.61 | 0.53 | $1.14 \times 10^{-11}$ |
| | RDFU | 1.42 | 0.65 | 0.03 |
| | WOF | -0.96 | 0.58 | 0.14 |
| | FFU × SN | 1.34 | 0.65 | 0.04 |
| | RDRFU × SN | -0.77 | 0.92 | 0.40 |
| | WOF × SN | -1.21 | 0.93 | 0.19 |

ns indicates the increase or decrease was not significant over the intercept value (Farmer fertilizer use under current plant density).

season (Table 2). The yield of the crop that was grown without fertilizer was not affected by season in both study locations. The maize yield reduction resulting from WOF was significant (p = 0.003) in CRV whereas the yield reduction associated with WOF treatment was not significant (p = 0.14) in Jimma as compared to the FFU.

### The effect of fertilizer and plant density on maize yield

Maize yield estimates with fertilizer use, season number, and plant density are presented in Table 3. The variation in maize yield was more explained when plant density was included in the model as a fixed effect factor in addition to fertilizer treatment and season number in both locations. Akaike information criterion (AIC) in the original model was 419.3 whereas it was 411.6 in the improved model in CRV. The improved model was significantly ($\chi 2_{(6)} = 19$; p = 0.003) better than the original model in CRV. In Jimma, the AIC was reduced from 529.9 in the original model to 492.9 in the improved model and this improved model was significantly better ($\chi 2_{(6)} = 49$; p = $2.7 \times 10^{-9}$) than the original model.

In CRV, the highest maize grain yield (4.5 t ha$^{-1}$) was achieved from the FFU and RDFU when the crop was grown at 53,333 plants ha$^{-1}$ plant density whereas the lowest grain yield (2.6 t ha$^{-1}$) was obtained from control treatment of 32,443 plants ha$^{-1}$. In both plant densities, farmer's fertilizer use (FFU) and redesigned fertilizer use (RDFU) did not result in significantly different maize yields in CRV but RDFU significantly increased maize grain yield over the WOF-density and FFU-density combinations in Jimma when the crop was grown with redesigned plant density (62,000 plants ha$^{-1}$). The highest maize grain yield (8.0 t ha$^{-1}$) was achieved from the combination of RDFU and redesigned density whereas the lowest yield (3.0 t ha$^{-1}$) was obtained from the combination of control and current plant density (27,724 plants ha$^{-1}$) in Jimma (Table 4). In this region, the average maize grain yield when the crop was grown without fertilizer and redesigned plant density was 4.3 t ha$^{-1}$. In the Jimma study area, maize grown with FFU and plant density resulted in 6.0 t ha$^{-1}$ which was not significantly higher than the control yield (4.3 t ha$^{-1}$) under the same plant density. The estimated maize yield for FFU and RDFU was the same for redesigned plant density in CRV whereas the yield of the crop was the same for these fertilizer treatments under current plant density in Jimma (Table 4).

**Table 3. Mixed model parameter estimates with standard errors in brackets of fertilizer use over two seasons of maize grain yield (t ha$^{-1}$) at 15% moisture content in CRV and Jimma, Ethiopia.**

| Region | Parameter | Estimates (t/ha) | Stand Error | P value |
|---|---|---|---|---|
| CRV | Intercept (FFU yield in season 1) | 4.38 | 0.39 | $2 \times 10^{-16}$ |
| | RDFU | 0.68 | 0.51 | 0.22 |
| | WOF | -0.81 | 0.51 | 0.11 |
| | FFU × SN | -1.55 | 0.51 | $2.3 \times 10^{-4}$ |
| | Plant density | 1.66 | 0.51 | $1.4 \times 10^{-4}$ |
| | RDFU×SN | -0.95 | 0.72 | 0.18 |
| | WOF×SN | -0.36 | 0.72 | 0.61 |
| | RDFU × plant density | -0.28 | 0.72 | 0.69 |
| | WOF × plant density | -0.78 | 0.72 | 0.27 |
| | SN × plant density | -1.58 | 0.72 | 0.02 |
| | RDFU × SN × plant density | 0.39 | 1.01 | 0.70 |
| | WOF × SN × plant density | 0.95 | 1.01 | 0.35 |
| Jimma | Intercept (FFU yield in season 1) | 3.74 | 0.57 | $7.5 \times 10^{-9}$ |
| | RDFU | 0.50 | 0.72 | 0.48 |
| | WOF | -0.58 | 0.72 | 0.42 |
| | FFU × SN | 1.68 | 0.72 | 0.02 |
| | Plant density | 1.74 | 0.72 | 0.02 |
| | RDFU × SN | -0.96 | 1.01 | 0.34 |
| | WOF × SN | -1.88 | 1.03 | 0.06 |
| | RDFU × plant density | 1.84 | 1.01 | 0.07 |
| | WOF × plant density | -0.77 | 1.01 | 0.44 |
| | SN × plant density | -0.60 | 1.01 | 0.55 |
| | RDFU × SN × plant density | 0.38 | 1.44 | 0.79 |
| | WOF × SN × plant density | 1.24 | 1.44 | 0.39 |

RDFU, FFU, WOF represent redesigned fertilizer use, farmer's fertilizer use and without fertilizer whereas SN stands season number.

In CRV, FFU and RDFU resulted in 38% and 42% more yield compared to the control treatments when maize was planted at 32,443 plants ha$^{-1}$ whereas the yield advantage of FFU and RDFU was 40% for each fertilizer use at 53,333 plants ha$^{-1}$. In Jimma, the FFU and the RDFU resulted in 53% more maize grain yield than the maize grown without fertilizer when the crop was grown at 27,724 plants ha$^{-1}$. In the same region, the yield gains of FFU and RDFU fertilizer treatments over the control increased to 39% and 86% with the redesigned plant density (62,000 plants ha$^{-1}$). In addition, the yield gain of RDFU surpasses the yield gain of FFU by 47% under redesigned plant density in Jimma.

An interaction effect of fertilizer use and plant density resulted in a significant maize yield increase in Jimma whereas in CRV, the yield increase due to the interaction effect between these factors was not significant. Redesigned plant density significantly (p<0.001) increased maize yield compared to the current plant density in CRV. However, RDFU did not result in a significant yield gain over the FFU in the same region (Table 5) but both FFU and RDFU significantly increased yield over the control treatment in this study area.

## Internal nutrient use efficiency

The internal nutrient use efficiency of maize in CRV and Jimma is presented in Table 6. Nitrogen use efficiency (NUE), phosphorus use efficiency (PUE), and potassium use efficiency (KUE) of maize were not significantly affected by fertilizer use treatments in CRV. A similar

**Table 4. Pairwise comparison of mean grain yield (t ha$^{-1}$) of maize at 15% moisture content for fertilizer use-plant density combinations in CRV and Jimma.**

| Region | Plant density (plant ha$^{-1}$) | Fertilizer use | | |
|---|---|---|---|---|
| | | **WOF** | **FFU** | **RDRF** |
| CRV | 32,443 | 2.6$^a$ | 3.6a$^{bc}$ | 3.7b$^c$ |
| | 53,333 | 3.2$^{ab}$ | 4.5$^c$ | 4.5$^c$ |
| | p value < 0.001 | | | |
| Jimma | 27,724 | 3.0$^a$ | 4.6$^a$ | 4.6$^{ab}$ |
| | 62,000 | 4.3$^a$ | 6.0$^b$ | 8.0$^c$ |
| | p value <0.001 | | | |

Means followed with the same superscript/s within a region are not significantly different at 0.05 p level. WOF, FFU and RDFU represent without fertilizer, farmer's fertilizer use and redesigned fertilizer use respectively.

response was observed in Jimma for NUE and PUE but KUE was significantly affected by fertilizer use (Table 6). In this region, RDFU significantly increased KUE compared to the WOF treatment. Internal N, P, and K use efficiency of maize with RDFU was not significantly different from internal use efficiency of these nutrients of the crop when FFU was applied and the crop was grown without fertilizer in Jimma. However, in CRV, the internal use efficiency of N and K of the crop-grown WOF was significantly higher than the internal use efficiency of these nutrients when FFU was used. The internal use efficiency of N with RDFU and WOF treatments was similar to the N use efficiency of the crop when FFU was applied in CRV. RDFU resulted in 43% and 69% more NUE and PUE compared to the control whereas the FFU resulted in 27% and 41% more NUE and PUE in a respective order in Jimma (S3 File). The result also revealed that RDFU and FFU increased KUE by 60% and 81% respectively compared to the maize grown without fertilizer in the same region.

## Economic feasibility of fertilizer uses and plant density

All fertilizer treatments under both current and redesigned densities were economically feasible in both seasons in CRV (Fig 2 and S4 File). In Jimma however, current and redesigned fertilizer uses did not result in greater than 1 value-cost ratio under current plant density in the first cropping season (2017 season) showing that they were not profitable. Similar to CRV, the WOF, FFU, and RDFU fertilizer treatments resulted in a greater than 1 value-cost ratio and

**Table 5. Comparison of mean grain yield (t ha$^{-1}$) of maize at 15% moisture content for fertilizer use and plant density treatments in CRV.**

| Fertilizer treatments | Maize yield (t ha$^{-1}$) |
|---|---|
| WOF | 2.9$^a$ |
| FFU | 4.0$^b$ |
| RDFU | 4.1$^b$ |
| p- value | 3.9 ×10$^{-6}$ |
| Plant density (plants ha$^{-1}$) | |
| 32,443 | 3.3$^a$ |
| 53,333 | 4.1$^b$ |
| p-value | 0.001 |

Means followed with the same superscript/s are not significantly different at 0.05 probability level. WOF, FFU and RDFU represent without fertilizer, farmer's fertilizer use and redesigned fertilizer use respectively.

**Table 6. Mixed model parameter estimates with standard errors in brackets (SE) of N, P and K use efficiencies of maize in CRV and Jimma, Ethiopia.**

| Region | Nutrient | parameter | Estimates (kg yield/kg nutrient uptake) | Stand Error | P value |
|--------|----------|-----------|------------------------------------------|-------------|---------|
| CRV | N | Intercept (FFU) | 46.0 | 4.4 | $5.9 \times 10^{-8}$ |
| | | RDFU | 9.8 | 5.7 | 0.119 |
| | | WOF | 15.2 | 5.7 | 0.024 |
| | P | Intercept (FFU) | 326.0 | 39.6 | $9.4 \times 10^{-6}$ |
| | | RDFU | 24.4 | 39.5 | 0.551 |
| | | WOF | 70.2 | 39.5 | 0.106 |
| | K | Intercept (FFU) | 48.0 | 13.4 | 0.003 |
| | | RDFU | 28.8 | 18.1 | 0.142 |
| | | WOF | 42.4 | 18.1 | 0.041 |
| Jimma | N | Intercept (FFU) | 108.2 | 21.2 | $6 \times 10^{-4}$ |
| | | RDFU | 13.4 | 23.5 | 0.584 |
| | | WOF | -23.2 | 23.5 | 0.353 |
| | P | Intercept (FFU) | 643.2 | 157.2 | $1.8 \times 10^{-3}$ |
| | | RDFU | 129.4 | 196.8 | 0.529 |
| | | WOF | -186.4 | 196.8 | 0.371 |
| | K | Intercept (FFU) | 108.9 | 26.2 | $5.9 \times 10^{-3}$ |
| | | RDFU | 14.3 | 20.1 | 0.495 |
| | | WOF | -40.8 | 20.1 | 0.076 |

WOF, FFU and RDFU represent without fertilizer, farmer's fertilizer use and redesigned fertilizer use respectively. NUE, PUE and KUE are nitrogen use efficiency, phosphorus use efficiency and potassium use efficiency.

were profitable in the second season (2018 season) in Jimma. Likewise, increasing planting density from 32,443 plants ha$^{-1}$ to 53,333 plants ha$^{-1}$ resulted in a value-cost ratio of greater than 1 in the first season when maize was grown WOF, with the FFU and with RDFU treatments. Unlike in CRV, increasing plant density from 27,724 plants ha$^{-1}$ to 62,000 plants ha$^{-1}$ did not lead to a greater than 1 value-cost ratio under a WOF in the first season in Jimma. The result in the second cropping season was similar for CRV and Jimma, in which the redesigned plant density resulted in less than 1 value-cost ratio under FFU but in WOF and RDFU increasing density led to profitability in both seasons (Fig 3 and S4 File).

### Relation of yield to soil factors

Organic carbon, total N, and available P contributed much to the principal component one (PC-1) in Jimma, whereas exchangeable K and soil pH contributed much to the principal component two (PC-2) in the region (Fig 4 and S3 Fig). Soil pH highly related to maize yield in Jimma which was not the case in CRV (S2 Fig). In CRV, total soil N, soil pH, and soil organic carbon had much contribution to PC-1 whereas available (extractable) P and exchangeable K contributed much to PC-2. Maize yield was positively correlated with soil OC soil N and soil pH in Jimma (S5 File and S3 Fig). In CRV, OC and soil N content were positively related to maize yield (Fig 4 and S2 Fig).

## Discussion

### Fertilizer use and plant density in maize

Accounting planting density improved our model quality compared to the model in which only fertilizer use and season number were used. This means that using fertilizer, season number, and plant density as fixed factors and farms as random factors explains maize yield

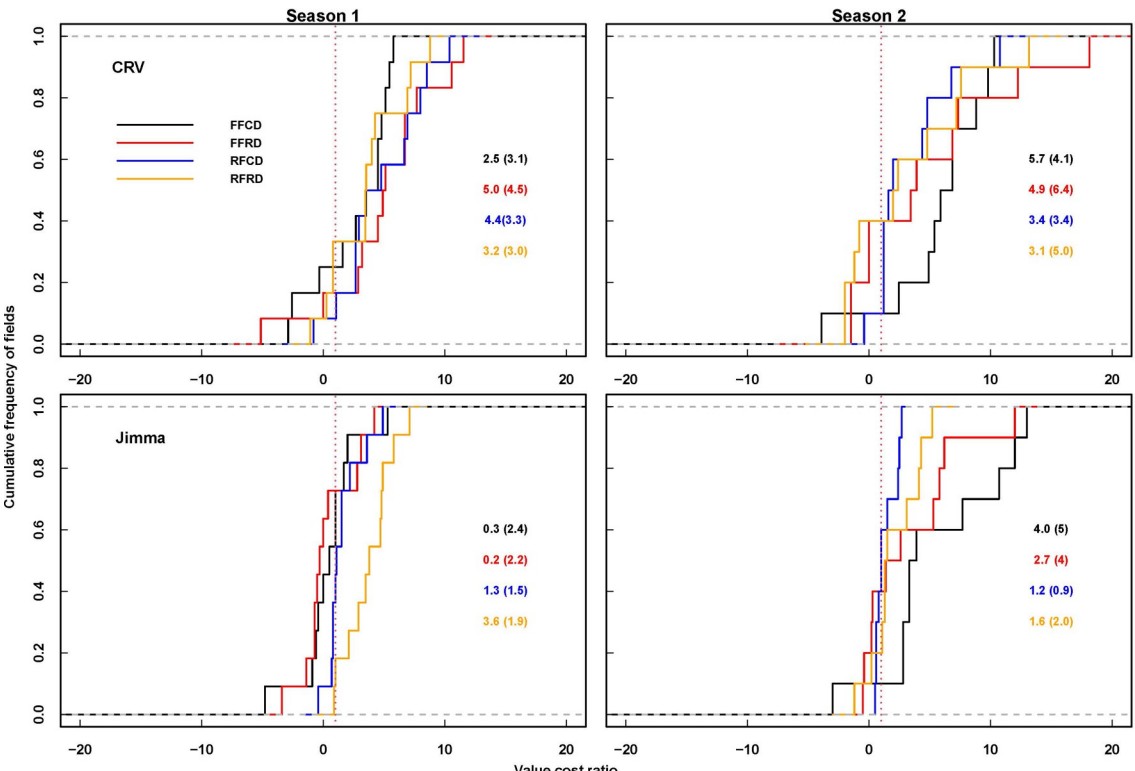

**Fig 2. Economic feasibility (value cost ratio) of fertilizer use in maize in CRV and Jimma during 2017 and 2018 cropping seasons.** FFCD, FFRD, RFCD and RFRD represent farmer's fertilizer use with current density, farmers fertilizer use with redesigned density, redesigned fertilizer with current density and redesigned fertilizer use with redesigned density respectively.

variability. The farmer's fertilizer uses (FFU) and redesigned fertilizer use (RDFU) significantly increased maize grain yield compared to the control in CRV. This study, however, did not show the superior yield from redesigned fertilizer use compared to the farmer's fertilizer use (FFU) in this region. However in Jimma, combinations of redesigned fertilizer use and redesigned plant density (62,000 plants ha$^{-1}$) significantly increased maize grain yield over other fertilizer-plant density treatment combinations (Table 4, S2 File and S1 Fig). This underlines the importance of redesigning fertilizer use rather than using the current farmers' use especially when plant density is increased [25, 26]. The result in general shows the importance of fertilizer use (FFU and redesigned) for improving maize yield and narrowing maize yield gaps in the regions. To attain a high yield level, the crop needs high nutrient input, and good husbandry regardless of the region. The maize yield with FFU significantly varied across seasons whereas the yield from RDFU was not dependent on season number in both study regions. This finding is in agreement with [27–29]. Here, we see the two-faced importance of redesigning fertilizer use (1) to improve productivity and (2) to increase yield stability over season. This finding complements [30] who investigated the balanced nutrition of maize with NPK and trace secondary-micro-nutrients to reduce yield variability.

Despite its inconsistent response across seasons, redesigning planting density resulted in a significant maize yield increase in CRV. The interaction between plant density and fertilizer use in this region did not significantly improve yield. As this region is characterized by moisture limitation for crops, the interaction between density fertilizers might be hindered. The yield gains of redesigned plant density (62,000 plant ha$^{-1}$) compared to the current density (27,724 plants ha$^{-1}$) were 43%, 30%, and 74% respectively when maize was grown with WOF,

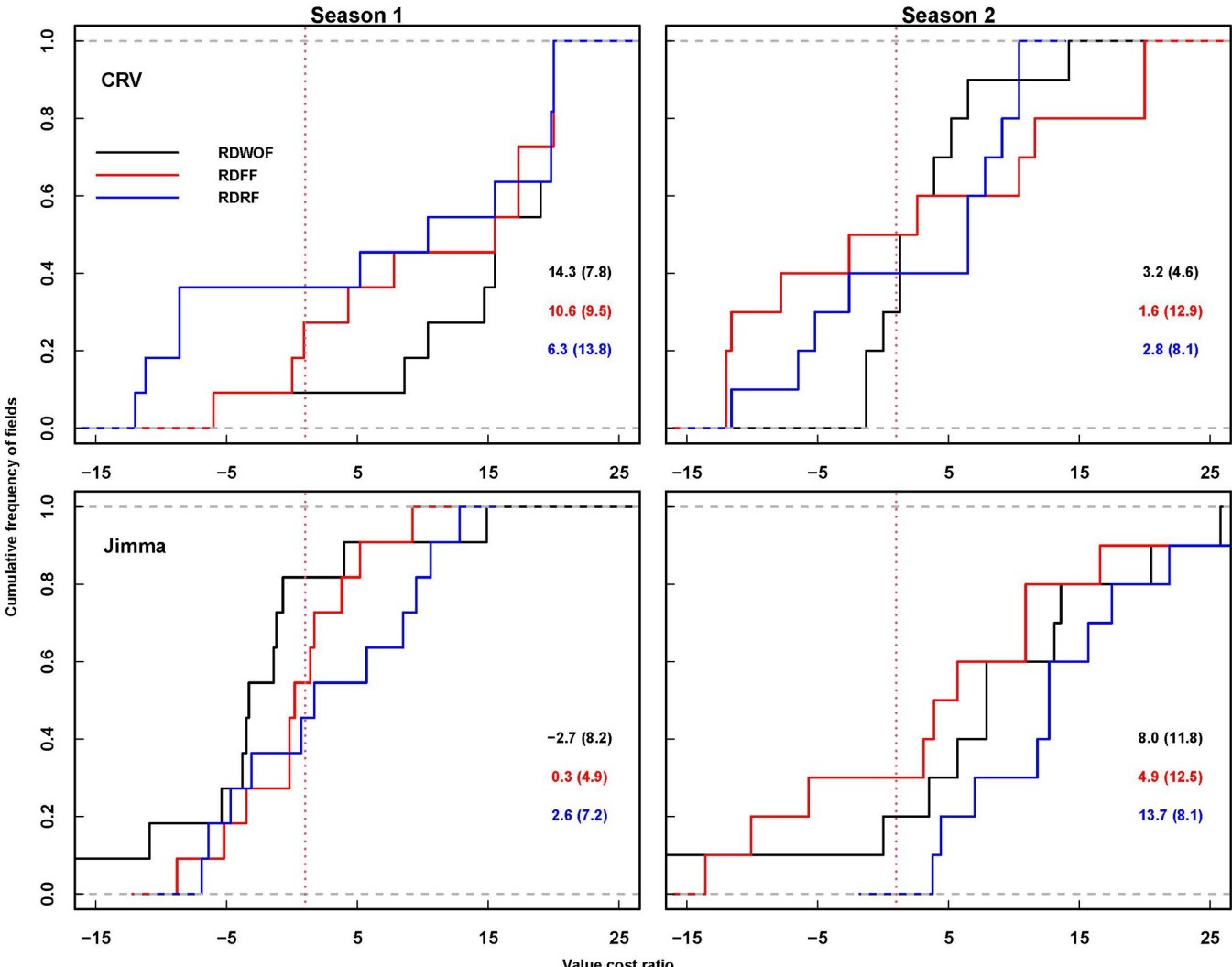

**Fig 3. Economic feasibility (value cost ratio) of plant density in maize in CRV and Jimma during 2017 and 2018 cropping seasons.** RDWOF, RDFF and RDRF represent redesigning density under control, redesigning density under farmer's fertilizer use and redesigning density under redesigned fertilizer use treatments respectively.

FFU, and RDFU fertilizer treatments in Jimma whereas in CRV the yield advantages of redesigned planting density (53,333 plants ha$^{-1}$) over the current plant density (32,443 plants ha$^{-1}$) were 23,%, 25%, and 23% respectively under the WOF, FFU, and RDFU fertilizer treatments with an average 24% yield advantage. The yield gains of redesigned plant density over the current practices were higher in Jimma compared to CRV under all fertilizer treatments. These findings highlight the importance of using higher planting density with high nutrient input in improving maize productivity in high maize potential regions like Jimma. Other studies for example, [31] affirmed that higher plant density leads to higher grain yield in some parts of the maize-growing area of Ethiopia.

In CRV, the soil is sandy and relatively not fertile compared to the soil in Jimma and therefore it is not logical to increase the plant density more than the recommended density which is 53,333 plants ha$^{-1}$. The current recommended (53,333 plants ha$^{-1}$) was found to be enough for the region. The current density (32,443 plant ha$^{-1}$) resulted in a significantly lower yield than

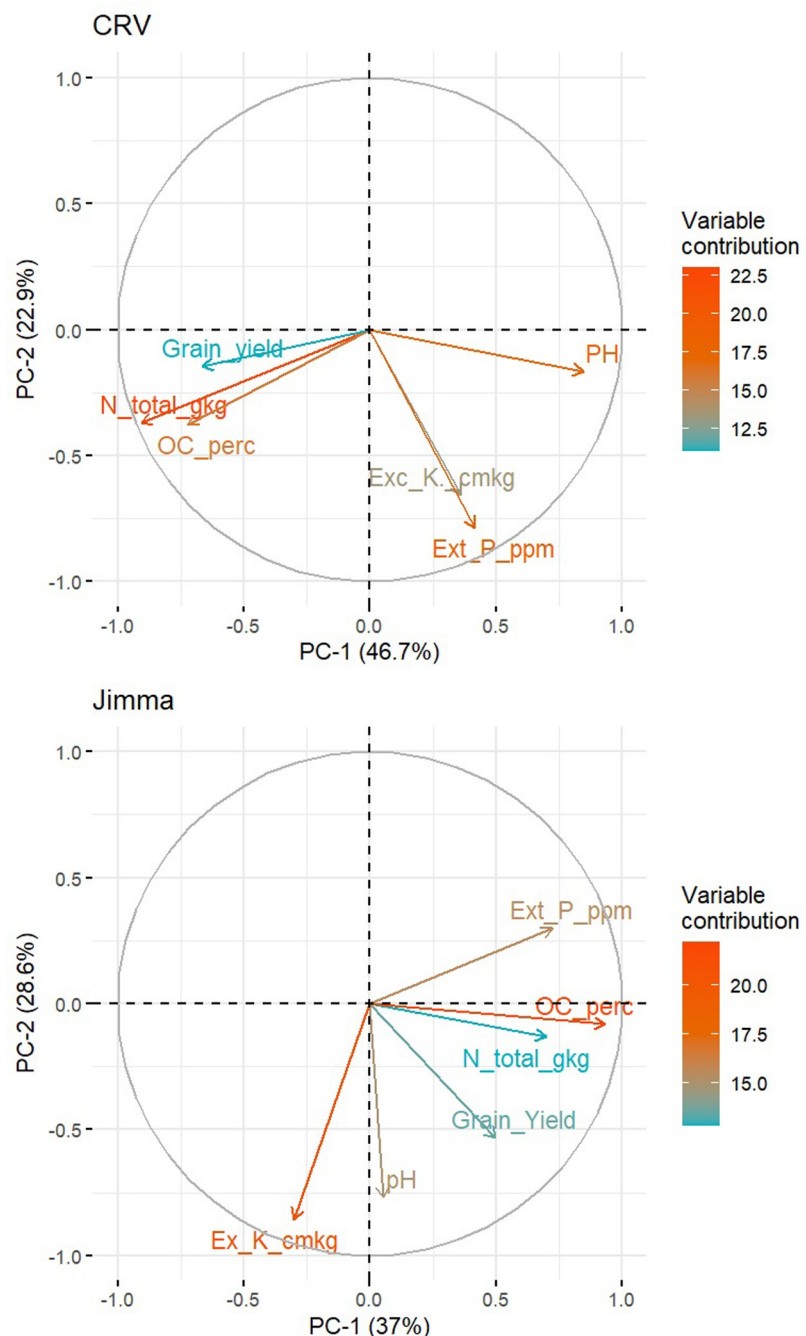

**Fig 4. Plots showing the relation between variables and their contribution to the first two principal components (PC-1 and PC-2) in central rift valley (CRV) and Jimma in Ethiopia.** Variables with a latent vector with a value close to one indicate that the variable makes a large contribution to the PCs. The percentage of variance explained by each principal component is presented on the axis labels in brackets.

the recommended density in both seasons. This highlights the need to demonstrate the yield increase by using higher plant density to farmers which perhaps leads to the effective adoption of the recommended planting density. Increasing density in Jimma was beneficial in improving yield and economic return for the maize growers in the region. In addition, our analysis

indicated that organic carbon, total soil N, and P content showed some tendency to correlation with maize yield in both regions (Fig 4). In addition, soil pH showed a positive association with yield in Jimma which was not the case in CRV where pH hardly related to maize yield. This is attributed to the acidic nature of Jimma soil and increasing its pH to a neutral level could also be accompanied by yield improvement. This agrees with [32] findings. The relation of soil parameters to crop yield is generally weak.

FFU and RDFU were profitable in maize production in both seasons in CRV when maize was planted at its current density and redesigned density whereas in Jimma these fertilizer uses were profitable when plant density was redesigned to 62,000 plants ha$^{-1}$. This shows that increasing plant density without increasing fertilizer input is not feasible in low fertile soil like soils in Jimma. This is because investment to maintain soil fertility for crop production in the region is very poor. Indeed some farms follow crop rotation principles while few farms leave crop residue as mulch on their fields during crop harvest and most farms do not employ soil and water conservation practices [33]. Furthermore, farms in this region do not practice applying manure to their fields [33]. Redesigning (increasing) plant density was associated with a positive return in CRV and in Jimma, it was highly profitable when nutrient input was optimized.

## Implications in narrowing yield gaps

On average, growing maize at redesigned planting densities with applications of the redesigned fertilizer uses in CRV and Jimma resulted in 72% and 51% of the potential yield of maize suggesting the importance of redesigning the plant density and fertilizer use for narrowing maize yield gaps in both regions. The lower proportion (relative to Yw) of the achieved yield in Jimma compared to CRV is indeed due to the high yield potential yield of maize in Jimma (15.7 t ha$^{-1}$) compared to the potential yield (6.2 t ha$^{-1}$) of the crop in CRV. This indicates that there is a possibility to further improve maize yield and narrow the yield gap in the Jimma area with good agronomy. Either in combination where significant interaction between them is observed or separately, fertilizer use [19] and plant density plays a significant role in narrowing maize yield gaps. The result agrees with the previous findings [19, 34–36] highlighting the use of a modest NPK fertilizer rate could narrow the maize yield gap down to 50% × water-limited yield potential (YW) of the crop in CRV and southwestern maize growing parts of Ethiopia. Indeed, the fertilizer was redesigned to achieve the 50% of the water-limited yield potential. In CRV however, we see an achievement of 72% of Yw which is more than 50%. This might be on one hand due to enough rainfall received during the study period than what was used for potential yield estimation and on the other hand, it might be due to the use of better agronomic practices other than fertilizer and plant density. We do argue here that fertilizer and plant density are not the only yield determinants in crop production. This finding is supported by earlier findings for example [31] which argued that higher maize grain yield could be achieved from higher plant density in Ethiopia. We can elucidate that increasing plant density from current to recommended in CRV and to redesigned in Jimma is a cheap option for narrowing the maize yield gap in the regions with low cost for smallholder farmer level.

## Methodological consideration

At the onset of the on-farm trial, the assumption was to set fertilizer treatments to apply the deficit nutrient amounts from fertilizer as a certain amount can be supplied from soil [37]. Given the presence of many soil types with different fertility regimes in Ethiopia, soils in CRV can supply enough K whereas soils in Jimma can supply almost enough P for 50% potential yield of maize, and their values were set to 0 kg ha$^{-1}$ for the respective nutrient in the respective

regions in the RDFU fertilizer treatment. The values of these nutrients in RDFU fertilizer treatment resembled an omission trial set up [19, 34] which was not an omission trial but the nutrient requirement was optimized quantitatively for a given target yield accounting for the soil supply and nutrient input from fertilizer. However, we were aware of the variability of these nutrients supplied from soil across fields within a region set up [29, 38]. We also acknowledge that the assessment of planting density considers more levels where yield plateau could be visible to recommend optimal plant population [26, 31]. However, we focused on demonstrating the potential of higher plant density compared to what farmers are using to motivate and influence them towards the adoption of higher plant density in maize production.

## Conclusion

The study investigated the importance of fertilizer use and redesigning plant density in maize production for improving productivity and narrowing the yield gaps of the crop in CRV and Jimma. With redesigned fertilizer use, 66% of the potential yield (Yw) of maize was achieved in CRV, and in Jimma, 51% of the potential yield of the crop was achieved when maize was grown with a combination redesigned density (62,000plants ha$^{-1}$) and redesigned fertilizer use. This suggests that good agronomy such as redesigned fertilizer use and redesigned plant density help narrow maize yield gaps in CRV and Jimma regions of Ethiopia. Growing maize with redesigned fertilizer use leads to high nutrient use efficiencies and stable yield. Organic carbon and N content of soil are closely related to the grain yield of maize compared to other soil parameters. Based on the fertilizer cost and maize grain price during the study period, both current and redesigned fertilizer uses are economically feasible under both current and redesigned plant density in CRV whereas in Jimma redesigned fertilizer use is economical irrespective of the plant density. Planting maize in redesigned density (53,333 plants ha$^{-1}$) is economical when maize was grown without fertilizer, farmers fertilizer use and redesigned fertilizer use in CRV whereas redesigning planting density to 62,000 plants ha$^{-1}$ is economically feasible when fertilizer use is redesigned to optimal requirements in Jimma. Our investigation also revealed that in addition to its role in improving yield and nutrient use efficiency, redesigned fertilizer use reduced yield variability across seasons at both CRV and Jimma study locations.

## Supporting information

**S1 File. Data of precipitation, maximum temperature and minimum temperature of CRV and Jimma during maize growing seasons in 2017 and 2018.** DOY, T_max and T_min refer to day of the year, maximum temperature and minimum temperature.
(CSV)

**S2 File. Data of maize grain yield response to fertilizer use, plant density and farm class in CRV and Jimma in 2017 and 2018 main seasons.** WOF, FFU and RDFU stand for without fertilizer, farmer's fertilizer use and redesigned fertilizer use respectively.
(CSV)

**S3 File. Data of maize physiological nutrient use efficiency as a function of fertilizer use.** WOF, FFU and RDFU stand for without fertilizer, farmer's fertilizer use and redesigned fertilizer use respectively whereas NUE, PUE and KUE are N use efficiency, P use efficiency and K use efficiency.
(CSV)

**S4 File. Data of value cost ratio of maize crop management practices in CRV and Jimma in 2017 and 2018.** FFCD, FFRD, RFCR and RFRD represent farmer's fertilizer use under current

density, farmer's fertilizer use under redesigned density, redesigned fertilizer under current density and redesigned fertilizer under redesigned density respectively. RDC, RDFF and RDRF stands for redesigned density under control, redesigned density under farmer's fertilizer use and redesigned density under redesigned fertilizer.
(CSV)

**S5 File. Data of soil parameters and maize yield, in CRV and Jimma, Ethiopia.** Ex_K_cmkg refer to exchangeable K (cmole/kg), whereas ext_P_ppm is extractable P (ppm), N_tot_gkg is total nitrogen (g/kg) and OC_perc is orgnic carbon (%).
(CSV)

**S1 Fig. Boxplots of yield response of maize to plant density under different fertilizer uses, in CRV and Jimma, in Ethiopia.**
(EPS)

**S2 Fig. Relation of soil parameters and maize yield in CRV, Ethiopia.** TN and Avp refer to total nitrogen and available phosphorus.
(EPS)

**S3 Fig. Relation of soil parameters and maize yield in Jimma, Ethiopia.** Ex_K_cmkg refer to exchangeable K (cmole/kg), whereas ext_P_ppm is extractable P (ppm), N_tot_gkg is total nitrogen (g/kg) and OC_perc is orgnic carbon (%).
(EPS)

## Acknowledgments

This work was supported, in whole or in part, by the Bill & Melinda Gates Foundation [INV-008260] through the CIMMYT, Taking Maize Agronomy to Scale in Africa (TAMASA) project. We also acknowledge Pytrik Reidsma, Katrien Descheemaeker, and Martin van Ittersum for their unreserved contribution to and shaping the proposal of the study. The authors also acknowledge the development agents who assisted with the fieldwork and data collection and the farmers for their willingness to host and manage the fields during the experimental period.

## Author Contributions

**Conceptualization:** Workneh Bekere Kenea, Tesfaye Balemi.

**Data curation:** Workneh Bekere Kenea.

**Formal analysis:** Workneh Bekere Kenea.

**Funding acquisition:** Tesfaye Balemi.

**Methodology:** Workneh Bekere Kenea, Tesfaye Balemi, Amsalu Nebiyu.

**Project administration:** Tesfaye Balemi.

**Software:** Workneh Bekere Kenea.

**Supervision:** Tesfaye Balemi, Amsalu Nebiyu.

**Validation:** Amsalu Nebiyu.

**Visualization:** Workneh Bekere Kenea, Tesfaye Balemi, Amsalu Nebiyu.

**Writing – original draft:** Workneh Bekere Kenea.

**Writing – review & editing:** Workneh Bekere Kenea, Tesfaye Balemi, Amsalu Nebiyu.

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
