## [Decision Letter · Decision Letter 0]

13 Nov 2023

PONE-D-23-22722Redesigning and Validation of Fertilizer use in Maize for Variable Plant densities and Farm classes in Central Rift Valley and Jimma in EthiopiaPLOS ONE

Dear Dr. Kenea,

Thank you for submitting your manuscript to PLOS ONE. After careful consideration, we feel that it has merit but does not fully meet PLOS ONE’s publication criteria as it currently stands. Therefore, we invite you to submit a revised version of the manuscript that addresses the points raised during the review process.

We look forward to receiving your revised manuscript.

Kind regards,

Paulo H. Pagliari

Academic Editor

PLOS ONE

Journal Requirements:

Did you know that depositing data in a repository is associated with up to a 25% citation advantage (https://doi.org/10.1371/journal.pone.0230416)? If you’ve not already done so, consider depositing your raw data in a repository to ensure your work is read, appreciated and cited by the largest possible audience. You’ll also earn an Accessible Data icon on your published paper if you deposit your data in any participating repository (https://plos.org/open-science/open-data/#accessible-data).

4. Please note that PLOS ONE has specific guidelines on code sharing for submissions in which author-generated code underpins the findings in the manuscript. In these cases, all author-generated code must be made available without restrictions upon publication of the work.

Please review our guidelines at https://journals.plos.org/plosone/s/materials-and-software-sharing#loc-sharing-code and ensure that your code is shared in a way that follows best practice and facilitates reproducibility and reuse.

6. Thank you for stating the following financial disclosure: 

"The study and conclusion belongs to the authors, but not the funding agency."

7. Thank you for stating the following in the Acknowledgments Section of your manuscript: 

"The authors would like to acknowledge the financial support of CIMMYT through the Taking Maize Agronomy to Scale in Africa (TAMASA) project for conducting the on-farm experiments. We also thank Pytrik Reidsma, Katrien Descheemaeker, and Martin van Ittersum for their unreserved contribution to and shaping the proposal of the study. The authors also acknowledge the development agents who assisted with the fieldwork and data collection and the farmers for their willingness to host and manage the fields during the experimental period."

"The study and conclusion belongs to the authors, but not the funding agency."

8. We note that you have indicated that data from this study are available upon request. PLOS only allows data to be available upon request if there are legal or ethical restrictions on sharing data publicly. For more information on unacceptable data access restrictions, please see http://journals.plos.org/plosone/s/data-availability#loc-unacceptable-data-access-restrictions. 

9. We note that you have included the phrase “data not presented” in your manuscript. Unfortunately, this does not meet our data sharing requirements. PLOS does not permit references to inaccessible data. We require that authors provide all relevant data within the paper, Supporting Information files, or in an acceptable, public repository. Please add a citation to support this phrase or upload the data that corresponds with these findings to a stable repository (such as Figshare or Dryad) and provide and URLs, DOIs, or accession numbers that may be used to access these data. Or, if the data are not a core part of the research being presented in your study, we ask that you remove the phrase that refers to these data.

10. Please ensure that you refer to Figure 5 in your text as, if accepted, production will need this reference to link the reader to the figure.

**Additional Editor Comments:**

Please follow the comments provided the reviewers to prepared a revised version of your manuscript.

Reviewers' comments:

Reviewer's Responses to Questions

**Comments to the Author**

1. Is the manuscript technically sound, and do the data support the conclusions?

Reviewer #1: Partly

Reviewer #2: Partly

Reviewer #3: Yes

Reviewer #4: Partly

2. Has the statistical analysis been performed appropriately and rigorously? 

Reviewer #1: No

Reviewer #2: No

Reviewer #3: Yes

Reviewer #4: I Don't Know

3. Have the authors made all data underlying the findings in their manuscript fully available?

Reviewer #1: Yes

Reviewer #2: No

Reviewer #3: Yes

Reviewer #4: No

4. Is the manuscript presented in an intelligible fashion and written in standard English?

Reviewer #1: Yes

Reviewer #2: No

Reviewer #3: Yes

Reviewer #4: Yes

5. Review Comments to the Author

Reviewer #1: PONE-D-23-22722-Redesigning and Validation of Fertilizer Use in Maize for Variable Plant Densities and Farm Classes in Central Rift Valley and Jimma in Ethiopia

Maize is a staple crop in many African counties and its availability is a measure of food security and any research leading to maize productivity leads to positive contribution. However, there are key fundamental concerns to be addressed in this study: From the title, the authors want to validate the fertilizer use- what necessitated the need for validation? At the end of the breeding trials, recommendations are usually made on plant density for each variety and it is not clear why the need for undertaking a trial on densities at this stage (no justification in the document). For fertilizers, this is understandable since most cases there are blanket recommendations without consideration of regions and soil fertility. Moreover, the word “redesigning” should be clearly defined as this sounds quite semantic.

Abstract

Generally, the abstract looks quite long, though this depends with Journal instructions. The abstract should be succinct, gives picture of the work and captures key aspects. I expected to see the general problem that necessitated this study. From this study, there is no gap that the authors aimed to fill and I was also unable to pick out the novelty of the work.

Keywords: the key words do not relate to the title of the manuscript.

Specifically, there is need to respond to following concerns:

Lines 11-12: “Redesigning and testing fertilizer use and plant densities in maize are important for grain self-sufficiency and livelihood improvement of smallholder farmers” –

* I do not seem to understand how redesigning and testing fertilizer use and plant densities play role in maize grain self-sufficiency. Moreover, there is need to define the term “redesigning”. It may imply that there was “designing” already, and if that is the case, what is wrong with initial design to warrant the “redesigning”? Please make this come out.

The densities in (32,443 and 53,333 plants ha-1 in Central Rift Valley (CRV); 27724 and 62,000 plants ha-1 in Jimma):

*Why extreme comparisons in CRV? Why didn’t the authors just adopt similar densities for both sites? Naturally, studies for densities recommend use of at least three so that it is possible observe yield plateau with increase in densities in order to make final conclusion on optimal density. Any comment on this?

Lines 14-15 “three farm classes (poor, medium and rich) in contrasting agro-ecologies of Ethiopia”-

*It is not clear whether the “farm classes” are part of experiment. Were these classes different in terms of soil nutrients levels?; Please clearly differentiate them.

Introduction

Lines 52-54 “Secondly, bringing more lands under cultivation is associated with increasing greenhouse gas emissions to the atmosphere which exacerbates climate change [6, 7]”-

*Can the authors explain how bringing more land under cultivation will exacerbate climate change. Otherwise can re-phrase the statement.

Lines 63-64 “The yield gap is the difference between the water-limited potential yield and the actual yield [9]”-

*Does it mean that yield gap can only be addressed through water management? What about use fertilizers and plant densities? Please comment on this.

Lines 67-70Accordingly, 12% more area (fields) were cultivated with maize in 2018 compared to the area occupied by the crop in 2014 whereas, the national average yield of maize increased from 3.4 t ha-1 in 2014 to 4 t ha-1 in 2018 [10, 11] which is equivalent to 18 % additional yield over the past four (2014-2018) years.

--Please re-look at the statement; from the statement, the increase in maize production was due to increase in area cultivated and yet authors talk of 18%additional yield; yield addresses the production per unit area; or productivity which was not the case in this scenario.

85-86 “The study also unraveled whether farms different resource endowment need different crop management technologies”.

*The statement needs to be rephrased” at this state, use of the word “whether” is wanting.

Materials and Methods

Lines 96-99 “The minimum temperature ranges from 5°C to 16.8°C in CRV whereas this temperature ranges from 5°C to 17 in Jimma (Fig. 1). The maximum temperature in CRV and Jimma ranges from 18°C to 31°C and from 17°C to 32°C respectively”

*Please re-look at the statements. The maximum or minimum temperature cannot be a range.

Lines 101-102 “Fig 1. Daily rainfall, minimum temperature and maximum temperature of CRV and Jimma in 2017 and 2018 years”

• The figure legends are missing and the caption not there. Make sure there are clear legends and the caption should be self- explanatory. The way it is currently, it conveys minimal information.

Lines 104-105 “Average fertilizer use and plant density of maize in the CRV and Jimma regions of Ethiopia are presented in Table 1”

• From table 1; there are rates of NPK as 0, 0, 0. Then current; 21.5, 12.8, 0.0 then redesigned: 40.8, 0.0 12.2 – These are very confusing since in one instance, K is omitted and the in the case of “redesigned scenario” it is the P that is omitted (for CRV) but for Jimma, the current is 53, 30, 0.0 and “redesigned” is 149.8, 9.0; 130.6…There is no clear explanation for this kind of adhoc applications. The omission/substation experiment are always systematic and should have been for both sites. Surprisingly, in the table I did not see the farm class factors (poor, medium and rich): were this in terms of soil nutrient status?. And if economic status; then did this have bearing on amount of NPK supplied? If yes, then how do you handle the 3 levels of fertilizers and the interaction with farm factors?: In general, the treatments in this table should be explained carefully with justification for randomness of treatment. This kind of applications lead to numerous and complex interactions that cannot be interpreted and hence makes the whole research futile. Interestingly enough, the tables/figures presented were quite simple and do not wholly represent treatments presented in table 1.

Line 114: 2.3Nutrient amounts in redesigned fertilizer use

• I guess this section is meant to provide methodology for part of objective 1:i.e to redesign and test fertilizer use and plant density under on-farm conditions and unravel their effect on maize yield and yield gaps-

• The authors are not clear on whether this was nutrient omission trial but they make mention of this, if so why do omission trial at this stage? However, based on table 1, if the omission trials were made, then other nutrient rates should have been held constant at every stage.

• Then there is also mention of yield gap issues – normally this is done through modelling but the questions is, why is it based on water-limitation and not nutrient? Why set it at 50%? And even in this instance, based on what plant density?

-Generally, the whole contents of this section do not relate to the title of the manuscript.

• Lines 135-136 “In CRV, 136 12 farms (4 poor, 4 medium and 4 rich) were selected”- This statement is not clear at all. Does wealth status have influence on soil fertility or purchasing powers? And if so, to what extent?: Please have a table and show all variations in input availability and nutrient levels due to wealth status of farms- e.g. livestock, chicken etc. and show how this affected, say the nutrient status – Honestly, I do not understand why these aspects were brought up. They could have been part of a larger project but I do not think they are necessary in this manuscript.

Line 169 .2.6 internal nutrient use efficiency

It is not clear why this has popped up. It was not mentioned under “introduction section” and I seem not to relate it with any of the objectives. This section should clearly show the formulae computation of these efficiencies. There are several of these efficiencies, and it is not clear how the authors settled on what is presented in table 3.

Line 215: Results

There is no need of presenting ANOVA while presenting the findings (Table 2). Instead take this to supplementary section.

After Table 2, there are no line numbering hence difficult to write report.

Nevertheless, section 3.2 Yield responses to fertilizer use, plant density and farm class, the results presented in table 2 “Coefficient of variation (%) of maize yield in CRV and Jimma areas under various fertilizer uses and plant densities” do you think this adds any value to your work? Some of the CV values are as high as 60. How important is this? The CV interpretation depends on sensitivity of the work and authors’ experience with the work. I not see this coming out in the discussion section.

Note there was already another table 2 (please label appropriately).

In this table, abbreviations such as WOF, FFU, and RDFU are not explained yet they are not universal. These should be clearly indicated in the table footnotes. Again, it seems the data were not analysed and hence not possible to tell if there were statistical differences between the treatments. The authors also make reference of figure 3. Unfortunately figures are not labelled. Without labelling the figures, it becomes difficult for readers/reviewers to understand the results.

Table 3- data not analysed. The authors provide maximum and minimum values instead of providing mean values. The means should then be compared amongst the treatments by having letter superscripts. These are lacking from this table

Please make sure all the figures are well labelled and captions indicated – the captions should be self- explanatory.

Due to the difficulties in interpreting the figures and lack of data analysis, the results need to clearly done. It is only then that discussions and subsequent sections can be meaningful. Hence I have not dedicated more time on reading the discussion.

Unless the result sections is seriously worked on, I am sorry so reject this work.

Reviewer #2: Generally, I would like to appreciate the effort made to collect and analyze data from the field and laboratory analysis

However, I have the following major concerns that makes it difficult to accept the MS in its current shape and argument:

In the abstract (and perhaps in the result: “Increasing plant density from current to 53,333 plants ha-1 in CRV and to 62,000 plants ha-1 in Jimma led to an abrupt maize yield improvement in both regions.” Attributing the yield differences obtained due to difference in locations (that includes differences in soil, water, nutrient, varieties and many other factors) as a difference occurred due to plant density is not justifiable.

Line 104-108: the statement “They were obtained from a farm survey conducted by (CIMMYT ca. Taking Maize Agronomy to Scale in Africa project) and the Ethiopian Institute of Agricultural Research (EIAR) was an average fertilizer use of the farmers in the region. “ Not clear”

The target yield was set at 50% of Yw in CRV for both seasons. In Jimma, 50% of water-limited potential yield was used as target yield in 2017 whereas in 2018, 50 and 70% of Yw were used as target yields.This is not clear. What is the justification for picking the target for redesigning fertilizer use at 50% of the water limited yield while the authors mentioned in the background as “These low yields have resulted from sub-optimal input use as a result of the economic constraint of the farms. Studies indicated that the cereal yield gap in Ethiopia is high and the country can be cereal self-sufficient by closing the yield gaps (to 80% of Yw) of the crop without import and area expansion”? The redesigned amount in the treatments could have been different if this targeting is different from 50%. Thus the manuscript doesn’t provide strong foundation for choosing the fertilizer amount leading to 50% of the water limited yield is the best and justifiable target.

Authors mentioned that water limited yield is estimated based on the result yield gap atlas as mentioned in line 118-119: “The target yields were obtained from Global Yield Gap and Water Productivity Atlas [18].” The scale of this information may not be applicable to differentiate the different farm categories anticipated by the authors. Moreover, the target yield is used in this MS to determine fertilizer rate that is considered as treatments, which is different from fertilizer use. There is no clear description as to how the rates are related to fertilizer use which have additional drivers such as access, capacity of the farmer to buy inputs, awareness and so on.

In line 116-118, authors mentioned as “The target yield was set at 50% of Yw in CRV for both seasons. In Jimma, 50% of water-limited potential yield was used as target yield in 2017 whereas in 2018, 50 and 70% of Yw were used as target yields.” Where does this 70 % assumption comes from? Why it refer to only Jima area only for 2018? In these sections and in several other sections, authors show inconsistencies in assumptions (changing across seasons in this case, and across locations, e.g plot size differences in two locations(I assume they have reported plot size differences in line145-146 because it matters), and differences in participating farmers per location and farmer catagories). This makes difficult to trust the result would lead to strong conclusion.

Related to plant Density:

1. What is the basis for picking those plant densities as redesigned densities? How is this consideration of “redesigned density” different from the ordinary two level density experiment? i.e what is the redesigning concept introduced here?

2. What is the justification for setting the current plant population as higher in CRV (32,443 plants ha-1 ) where moisture is the most limiting factor compared to Jimma area (27,724 plants ha-1) where much amount of seasonal rainfall is received and long maturing varieties are possible?

3. Often mentioned as intermediate density. What does it mean and how does this relate to redesigning?

Line 145-146: The plot sizes in CRV and Jimma were 6 m × 6 m and 6.0 m × 6.4 m respectively.

What is the reason for difference in size of plot between two locations?

What is the rationale for classifying the farmers in to three categories when the treatments used are same i.e. the redesign of fertilizer use is based on targeting 50% of the water limited yield

The various analysis undertaken are not well connected to answer the issue of redesigning although they may be good indicators for telling other stories for other purposes.

Coefficient of variability is not only generally high, but also mostly higher for the redesigned densities and in some cases (years) in redesigned fertilizers. What is the implication?

In the discussion part (line number and page number is not from start of result section) authors describe as: “The redesigned fertilizer use increased maize grain yield compared to the current fertilizer use in CRV and Jimma. Moreover, the redesigned fertilizer use significantly increased the grain yield and this result shows the importance of redesigning fertilizer use for improving maize yield and narrowing maize yield gaps in the regions.” Whereas the result presented in line 25 read as “….the yield advantage is not significantly higher compared to the farmers’ fertilizer uses in both regions.” This claim on the discussion is, therefore, not evidence based.

Similarly, in the discussion section it is written as : Redesigning planting density led to a significant maize yield increase in both seasons.” Whereas abstract authors mentiond as “In CRV, plant density and farm class did not affect maize yield in 2018 …”

Authors claim the following: Averaging on 2017 and 2018 cropping seasons, the achieved yield level by using redesigned fertilizer uses were 4.1 t ha-1 and 6.1 CRV and Jimma, respectively resulting in 65% and 38% of the water-limited potential yield of the crop in the respective regions.” How you end up with these percentages while you designed inputs to attain 50% of the water limited yield?

The conclusion boldly claim: “The study investigated the importance of redesigning fertilizer use and plant density” whereas the result shows non-significant difference quit in a number of combinations”. How?

While year is not mentioned as major factor, several instances in the manuscript deal with comparison of years whereas there is no evidence on the relative advantage of the so called redesigned recommendation compared with currently available recommendations that are already provided through the extension service. It only tried to compare with farer's practice and zero fertilizer.

Reviewer #3: The manuscript entitled Redesigning and Validation of Fertilizer use in Maize for Variable Plant densities and Farm classes in Central Rift Valley and Jimma in Ethiopia has several important information to Africa. However, it is not completely clear why improve the plant density associated with fertilizer, especially in the Abstract and Introduction sections. I would like to first hear from authors about the the scientific basis of the experiment. In addition, text should be sent to an expert in English. Last but not the least, there are some parts with no number of lines what makes difficult to write the suggestions.

General comments:

L11-12: What is grain self-suffiency?

L12: smallholder?

Line 19: Standardize whether kg/ha or kg ha-1 will be used. Check it throughout the manuscript.

L20: Use square brackets when you have to use more than once parentheses.

Do the same above.

L44: Why did you choose 2010? Please bring data from at least 2020.

L50: Please send it to an English review language. Do it with the manuscript to avoid this type of error.

Table 1. Write the meaning of abbreviation in the title of table. - Take care and check it in all tables and figures, it must be improved.

Table 2. Write the meaning of abbreviations.

Organic carbon and N content of soil are closely related to the grain yield of maize compared to other soil and socioeconomic factors. - Is this information important? Why was it not in the abstract?

Reviewer #4: The authors present a study on “redesigning” fertilizer application rate and planting density in two regions of Ethiopia: Central Rift Valley and Jimma. The paper is well written and of practical relevance for famers and the extension system.

There were some areas in the paper where I struggled to follow and I would suggest a revision:

Material and methods:

Line 119 – where did you get the information on nutrient content to compute physiological use efficiency?

Line 122 – how does the NOT relate to the trial sites? Can you expect the indigenous nutrient supply to be similar at the trial sites and the NOT?

Line 126: reference for the nutrient recovery efficiencies

Line 143/144: do you mean that one farm was one replicate?

Line 146: what is the area that was used for harvest? Full plot? Net plot? Size of net plot? How was the time for harvest determined?

Line 150: where does the intermediate density come from? Before you were talking about redesigned density or optimum.

Line 156: what was the maturity duration of the varieties?

Line 160: how old were the plants at the time of thinning?

What was the basis for the redesigned planting densities? You explain this for fertilizer, but not for the planting density.

Statistical analysis – this should be more comprehensive. You only explain how you compared treatments, but not how you established that there was a significant effect of a factor. Since you used several farms, did you use linear mixed modelling? Please provide the critical details. Did you perform statistical analysis for each region separately or in combination? Under results, you provide an ANOVA table that implies separate analysis. However figure 2 is only for Jimma – why did you omit CRV?

Internal nutrient use efficiency: sufficient to provide the reference, the details are not needed here.

Relation of maize yield to socio economic and soil factors. Am not an expert in the area, but nevertheless wondering if all the socio economic factors are meaningful here? Did you use any procedure to identify the best factors to be included in this analysis?

Line 208: what is the difference between experimental and sampled yield?

Line 210/211: first you use an average yield of 2017 and 2018, then you say you used the sampled yield of 2017. Please explain.

Discussion

You state that: “Moreover, the redesigned fertilizer use significantly increased the grain yield and this result shows the importance of redesigning fertilizer use for improving maize yield and narrowing maize yield gaps in the regions” Under results you wrote: “In both seasons, current and redesigned fertilizer uses did not result in significantly different maize yields in CRV and Jimma regions”. Please explain.

The interaction that you refer after this, is a singular event. How useful is this for your reasoning?

I suggest adding some background to the low planting densities that farmers use. Is there a risk associated with the higher densities? – maybe drought?

I would also suggest including a discussion of risk in the context of the cumulative frequency analysis: What % of farmers benefited? What % lost? What is acceptable? (it is not clear what the vertical line the corresponding figure represents – I have assumed 1)

6. PLOS authors have the option to publish the peer review history of their article (what does this mean?). If published, this will include your full peer review and any attached files.

Reviewer #1: **Yes: **Joseph Gweyi-Onyango

Reviewer #2: No

Reviewer #3: **Yes: **Letusa Momesso

Reviewer #4: No

---

## [Author Response · Author response to Decision Letter 0]

3 Jan 2024

We thank the reviewers for their generous comments that made a significant improvement of our manuscript. We have revised the manuscript to address the concerns raised during the evaluation and submitted.

---

## [Decision Letter · Decision Letter 1]

13 Feb 2024

PONE-D-23-22722R1Redesigning and Validation of Fertilizer use in Maize for Variable Plant densities and Farm classes in Central Rift Valley and Jimma in EthiopiaPLOS ONE

Dear Dr. Kenea,

Thank you for submitting your manuscript to PLOS ONE. After careful consideration, we feel that it has merit but does not fully meet PLOS ONE’s publication criteria as it currently stands. Therefore, we invite you to submit a revised version of the manuscript that addresses the points raised during the review process.

Please follow the instruction provided when preparing a revised document. One of the reviewers was not able to fully identify what had been changed from the prewvious file. Use word track changes when making changes to the document.

We look forward to receiving your revised manuscript.

Kind regards,

Paulo H. Pagliari

Academic Editor

PLOS ONE

Additional Editor Comments:

Please follow the instruction provided when preparing a revised document. One of the reviewers was not able to fully identify what had been changed from the prewvious file. Use word track changes when making changes to the document.

Reviewers' comments:

Reviewer's Responses to Questions

**Comments to the Author**

1. If the authors have adequately addressed your comments raised in a previous round of review and you feel that this manuscript is now acceptable for publication, you may indicate that here to bypass the “Comments to the Author” section, enter your conflict of interest statement in the “Confidential to Editor” section, and submit your "Accept" recommendation.

Reviewer #3: All comments have been addressed

Reviewer #4: (No Response)

2. Is the manuscript technically sound, and do the data support the conclusions?

Reviewer #3: Yes

Reviewer #4: Partly

3. Has the statistical analysis been performed appropriately and rigorously? 

Reviewer #3: Yes

Reviewer #4: I Don't Know

4. Have the authors made all data underlying the findings in their manuscript fully available?

Reviewer #3: Yes

Reviewer #4: Yes

5. Is the manuscript presented in an intelligible fashion and written in standard English?

Reviewer #3: Yes

Reviewer #4: Yes

6. Review Comments to the Author

Reviewer #3: (No Response)

Reviewer #4: Reviewer #4:

The responses by the authors are mainly fine but I am surprised that they have not modified their text accordingly (and if I have overlooked that – it would help to indicate in the response where the adjustment was made (line number)

Comment

Line 119 – where did you get the information on nutrient content to compute physiological use efficiency?

Response #24

We got the nutrient content from laboratory analysis for computing the physiological use efficiency. We computed uptake (kg/ha) and yield per uptake is a physiological use efficiency.

ReRe: this has not been included into the text, also not from which trials samples were analysed and used here.

Comment

Line 122 – how does the NOT relate to the trial sites? Can you expect the indigenous nutrient supply to be similar at the trial sites and the NOT?

Response #25

The NOT sites in the same location with this study and soil supply was estimated from the previous NOT trial.

ReRe: has the section with the NOTs been removed?

Comment

Line 143/144: do you mean that one farm was one replicate?

Response #26: Yes

ReRe: not included in text

Line 146: what is the area that was used for harvest? Full plot? Net plot? Size of net plot? How was the time for harvest determined?

Response #27

Harvesting was done from net plot size excluding the two border rows. The net plots were 6m ×4.8m in Jimma whereas in CRV the net plot size was 6m ×4.5m. Harvesting time was at physiological maturity when grain moisture was dropped to 15- 30% in most cases. This can be visually observed as leaves of the crop dries, cobs turns downwards from upright position.

ReRe: not included in text

Comment

Line 150: where does the intermediate density come from? Before you were talking about redesigned density or optimum.

Response #28

To be frank, intermediate density was included to mean the redesigned density and to avoid confusion; it is now changed to redesigned density.

ReRe: OK

Comment

Line 156: what was the maturity duration of the varieties?

Response #29

The maturity date of BH-660 (that was used in Jimma) was 160 days whereas BH-540 (that was used in CRV) was 140 days.

ReRe: not included in text

Comment

Line 160: how old were the plants at the time of thinning?

Response #30

The age of maize seedlings at thinning was 30 -45 days.

ReRe: not included in text

Comment

What was the basis for the redesigned planting densities? You explain this for fertilizer, but not for the planting density.

Responses#31

The bases of redesigning plant density are demonstration trial and consult of senior agronomists.

ReRe: The response could be more elaborate and should have been included in the text

Comment

Statistical analysis – this should be more comprehensive. You only explain how you compared treatments, but not how you established that there was a significant effect of a factor. Since you used several farms, did you use linear mixed modelling? Please provide the critical details. Did you perform statistical analysis for each region separately or in combination? Under results, you provide an ANOVA table that implies separate analysis. However figure 2 is only for Jimma – why did you omit CRV?

Response #32

The statistical analysis was conducted separately. From ANOVA table, we see that the interaction effect of fertilizer use and Density was significant only for Jimma which lead to produce fig 2result. Because, the interaction between fertilizer use and density was not significant for CRV, it was not presented.

ReRe: still not clear if mixed modelling was used or how else the multilocational situation was addressed.

Comment

Relation of maize yield to socio economic and soil factors. Am not an expert in the area, but nevertheless wondering if all the socio economic factors are meaningful here? Did you use any procedure to identify the best factors to be included in this analysis?

Responses #33

Some factors were reduced and relevant factors were included under supplementary materials.

ReRe: A more comprehensive response would have been appreciated.

Comment

Line 208: what is the difference between experimental and sampled yield?

Response#34

Experimental yield is what is measured from experimental plot whereas sampled yield is what was measured from farmer’s field (average).

ReRe: still not clear from the text

Comment

Line 210/211: first you use an average yield of 2017 and 2018, then you say you used the sampled yield of 2017. Please explain.

Response #35

For experimental yield, an average yield of on farm experiment in 2017 and 2018 was used, and for sampled yield, the yield sampled in 2017 from farmers’ fields was used.

ReRe: OK, if above is clarified in the text

7. PLOS authors have the option to publish the peer review history of their article (what does this mean?). If published, this will include your full peer review and any attached files.

Reviewer #3: No

Reviewer #4: No

---

## [Author Response · Author response to Decision Letter 1]

26 Mar 2024

Major changes in this revision

1. Farm class is not considered as experimental factor. It is removed from all parts of the manuscript

2. Statistical analysis: it is re-conducted with linear mixed model using lme4 and LmerTest. All parts of the parts updated accordingly and interpreted accordingly.

3. Relation of maize yield with soil parameters: socioeconomic parameters and sampled yield are removed. Experimental yield is related with soil parameter (of the experimental fields).

4. Yield variability assessment based on CV category was removed.

5. Revision that were not included in the text during the previous revision are now included in the text.

PLOS ONE 

www.ariessys.com

From:em@editorialmanager.com

To:Workneh Bekere Kenea

Tue, Feb 13 at 10:58 PM

PONE-D-23-22722R1

Redesigning and Validation of Fertilizer use in Maize for Variable Plant densities and Farm classes in Central Rift Valley and Jimma in Ethiopia

PLOS ONE

Dear Dr. Kenea,

Thank you for submitting your manuscript to PLOS ONE. After careful consideration, we feel that it has merit but does not fully meet PLOS ONE’s publication criteria as it currently stands. Therefore, we invite you to submit a revised version of the manuscript that addresses the points raised during the review process.

Please follow the instruction provided when preparing a revised document. One of the reviewers was not able to fully identify what had been changed from the prewvious file. Use word track changes when making changes to the document.

We look forward to receiving your revised manuscript.

Kind regards,

Paulo H. Pagliari

Academic Editor

PLOS ONE

Additional Editor Comments:

Please follow the instruction provided when preparing a revised document. One of the reviewers was not able to fully identify what had been changed from the prewvious file. Use word track changes when making changes to the document.

Reviewers' comments:

Reviewer's Responses to Questions

Comments to the Author

1. If the authors have adequately addressed your comments raised in a previous round of review and you feel that this manuscript is now acceptable for publication, you may indicate that here to bypass the “Comments to the Author” section, enter your conflict of interest statement in the “Confidential to Editor” section, and submit your "Accept" recommendation.

Reviewer #3: All comments have been addressed

Reviewer #4: (No Response)

2. Is the manuscript technically sound, and do the data support the conclusions?

Reviewer #3: Yes

Reviewer #4: Partly

3. Has the statistical analysis been performed appropriately and rigorously?

Reviewer #3: Yes

Reviewer #4: I Don't Know

4. Have the authors made all data underlying the findings in their manuscript fully available?

Reviewer #3: Yes

Reviewer #4: Yes

5. Is the manuscript presented in an intelligible fashion and written in standard English?

Reviewer #3: Yes

Reviewer #4: Yes

6. Review Comments to the Author

Reviewer #3: (No Response)

Reviewer #4: Reviewer #4:

The responses by the authors are mainly fine but I am surprised that they have not modified their text accordingly (and if I have overlooked that – it would help to indicate in the response where the adjustment was made (line number)

Comment

Line 119 – where did you get the information on nutrient content to compute physiological use efficiency?

Response #24

We got the nutrient content from laboratory analysis for computing the physiological use efficiency. We computed uptake (kg/ha) and yield per uptake is a physiological use efficiency.

ReRe: this has not been included into the text, also not from which trials samples were analysed and used here.

Response of the authors:

ReReRe: Grain and straw samples were collected from the on-farm trial of the first season (2017) and the NPK content was analyzed at WUR. This nutrient concentration was used for physiological use efficiency. This text was included in the methodology line #166 in clean manuscript and at 197 line # in track change doc.

Comment

Line 122 – how does the NOT relate to the trial sites? Can you expect the indigenous nutrient supply to be similar at the trial sites and the NOT?

Response #25

The NOT sites in the same location with this study and soil supply was estimated from the previous NOT trial.

ReRe: has the section with the NOTs been removed?

Response of the authors:

ReReRe: This fertilizer use treatments are the estimated NPK for achieving the 50% yield potential of the crop. To make it clear and how it differs from NOT, explanation has been given under discussion with a title of methodological consideration. See line# 433-446 on the clean paper and 609-623 line # on the track change doc. 

Comment

Line 143/144: do you mean that one farm was one replicate?

Response #26: Yes

ReRe: not included in text

Response of the authors 

ReReRe: Now it is included in the text at Line #132 on clean paper and line # 56-57 on track change.

Line 146: what is the area that was used for harvest? Full plot? Net plot? Size of net plot? How was the time for harvest determined?

Response #27

Harvesting was done from net plot size excluding the two border rows. The net plots were 6m ×4.8m in Jimma whereas in CRV the net plot size was 6m ×4.5m. Harvesting time was at physiological maturity when grain moisture was dropped to 15- 30% in most cases. This can be visually observed as leaves of the crop dries, cobs turns downwards from upright position.

ReRe: not included in text

Response: 

ReReRe: These responses are included in the text. The line numbers are 149-153 on the clean document and 174-178 line numbers on the track change. 

Comment

Line 150: where does the intermediate density come from? Before you were talking about redesigned density or optimum.

Response #28

To be frank, intermediate density was included to mean the redesigned density and to avoid confusion; it is now changed to redesigned density.

ReRe: OK

Comment

Line 156: what was the maturity duration of the varieties?

Response #29

The maturity date of BH-660 (that was used in Jimma) was 160 days whereas BH-540 (that was used in CRV) was 140 days.

ReRe: not included in text

Response:

ReReRe: The maturity date of BH-660 (that was used in Jimma) was 160 days whereas BH-540 (that was used in CRV) was 140 days. The statement was included in the paper in line number 145-146 on clean paper and 170-171 line numbers on the track change doc.

Comment

Line 160: how old were the plants at the time of thinning?

Response #30

The age of maize seedlings at thinning was 30 -45 days.

ReRe: not included in text

The age at which thinning was done was 30 -45 days after emergency and now included in the text. Line # 147-148.

Comment

What was the basis for the redesigned planting densities? You explain this for fertilizer, but not for the planting density.

Responses#31

The bases of redesigning plant density are demonstration trial and consult of senior agronomists.

ReRe: The response could be more elaborate and should have been included in the text

Response of the authors

ReReRe: The basis of redesigned plant density (CRV=53,333 plants ha-1 and 62,000 plants ha-1) is the performances in previous demonstration trials and consultation of senior agronomists. Soil resources and climatic elements (rainfall) of the regions were taken into account. These are included in the text at line numbers 121-224 in clean paper and136-140 line numbers in tack change doc.

Comment

Statistical analysis – this should be more comprehensive. You only explain how you compared treatments, but not how you established that there was a significant effect of a factor. Since you used several farms, did you use linear mixed modelling? Please provide the critical details. Did you perform statistical analysis for each region separately or in combination? Under results, you provide an ANOVA table that implies separate analysis. However figure 2 is only for Jimma – why did you omit CRV?

Response #32

The statistical analysis was conducted separately. From ANOVA table, we see that the interaction effect of fertilizer use and Density was significant only for Jimma which lead to produce fig 2result. Because, the interaction between fertilizer use and density was not significant for CRV, it was not presented.

ReRe: still not clear if mixed modelling was used or how else the multilocational situation was addressed.

Response of the authors:

ReReRe: In the version, linear mixed model was used for the statistical analysis replacing the factorial anova in the previous version. The result and other parts of the paper were updated accordingly. This was included in 155-163 line numbers on the clean doc and180-188 line numbers in the track change.

Comment

Relation of maize yield to socio economic and soil factors. Am not an expert in the area, but nevertheless wondering if all the socio economic factors are meaningful here? Did you use any procedure to identify the best factors to be included in this analysis?

Responses #33

Some factors were reduced and relevant factors were included under supplementary materials.

ReRe: A more comprehensive response would have been appreciated.

Response of the authors

ReReRe: In this revised version, socio-economic parameters (TLU, N use, P use, cropped area, maize area) and sampled yield are removed. Only the relation of experimental yield and soil parameter (of the experimental fields) were assessed.

Comment

Line 208: what is the difference between experimental and sampled yield?

Response#34

Experimental yield is what is measured from experimental plot whereas sampled yield is what was measured from farmer’s field (average).

ReRe: still not clear from the text

Response of the authors,

ReReRe: In this version, only one maize yield exists, the experimental yield. Sampled yield is removed.

Comment

Line 210/211: first you use an average yield of 2017 and 2018, then you say you used the sampled yield of 2017. Please explain.

Response #35

For experimental yield, an average yield of on farm experiment in 2017 and 2018 was used, and for sampled yield, the yield sampled in 2017 from farmers’ fields was used.

ReRe: OK, if above is clarified in the text

Response of the authors: 

ReReRe: the experimental yield that was related with the soil factors is the average yield of 2017and 2018 season grown under NPK (redesigned fertilizer use). Under NPK (redesigned fertilizer use) we assume that there was no yield limitation from nutrient perspective. Now there is no sampled yield in the paper and it might be clear. See line 203-206 of the paper (clean paper) and 468-488on track change. See also fig 4.

---

## [Decision Letter · Decision Letter 2]

6 May 2024

Redesigning and Validation of Fertilizer use in Maize for Variable Plant densities and Farm classes in Central Rift Valley and Jimma in Ethiopia

PONE-D-23-22722R2

Dear Dr. Kenea,

We’re pleased to inform you that your manuscript has been judged scientifically suitable for publication and will be formally accepted for publication once it meets all outstanding technical requirements.

Kind regards,

Paulo H. Pagliari

Academic Editor

PLOS ONE

Additional Editor Comments (optional):

Reviewers' comments:

Reviewer's Responses to Questions

**Comments to the Author**

1. If the authors have adequately addressed your comments raised in a previous round of review and you feel that this manuscript is now acceptable for publication, you may indicate that here to bypass the “Comments to the Author” section, enter your conflict of interest statement in the “Confidential to Editor” section, and submit your "Accept" recommendation.

Reviewer #4: All comments have been addressed

2. Is the manuscript technically sound, and do the data support the conclusions?

Reviewer #4: (No Response)

3. Has the statistical analysis been performed appropriately and rigorously? 

Reviewer #4: I Don't Know

4. Have the authors made all data underlying the findings in their manuscript fully available?

Reviewer #4: Yes

5. Is the manuscript presented in an intelligible fashion and written in standard English?

Reviewer #4: Yes

6. Review Comments to the Author

Reviewer #4: Thanks for the elaborated responses and detailed references to the revised versions of the manuscript. This was very helpful.

7. PLOS authors have the option to publish the peer review history of their article (what does this mean?). If published, this will include your full peer review and any attached files.

Reviewer #4: No

---

## [Editor Report · Acceptance letter]

28 May 2024

PONE-D-23-22722R2 

PLOS ONE

Dear Dr. Kenea, 

I'm pleased to inform you that your manuscript has been deemed suitable for publication in PLOS ONE. Congratulations! Your manuscript is now being handed over to our production team.

Kind regards, 

on behalf of

Dr. Paulo H. Pagliari 

Academic Editor

PLOS ONE